# It's DONE: Direct ONE-shot learning with Hebbian weight imprinting

## Abstract

Learning a new concept from one example is a superior function of the human
brain and it is drawing attention in the field of machine learning as a one-shot
learning task. In this paper, we propose one of the simplest methods for this task
with a nonparametric weight imprinting, named Direct ONE-shot learning (DONE).
DONE adds new classes to a pretrained deep neural network (DNN) classifier with
neither training optimization nor pretrained-DNN modification. DONE is inspired
by Hebbian theory and directly uses the neural activity input of the final dense
layer obtained from data that belongs to the new additional class as the synaptic
weight with a newly-provided-output neuron for the new class, by transforming all
statistical properties of the neural activity into those of synaptic weight. DONE
requires just one inference for learning a new concept and its procedure is sim-
ple, deterministic, not requiring parameter tuning and hyperparameters. DONE
overcomes a problem of existing weight imprinting methods that interfere with the
classification of original-class images. The performance of DONE depends entirely
on the pretrained DNN model used as a backbone model, and we confirmed that
DONE with current well-trained backbone models perform at a decent accuracy.

## 1   Introduction

As is well known, artificial neural networks are initially inspired by the biological neural network in
the animal brain. Subsequently, Deep Neural Networks (DNNs) achieved great success in computer
vision [9, 14, 20] and other machine learning fields. However, there are lots of tasks that are easy
for humans but difficult for current DNNs. One-shot learning is considered as one of those kinds
of tasks [5, 17, 19, 22, 27]. Humans can add a new class to their large knowledge from only one
input image but it is difficult for DNNs unless another specific optimization is added. Usually,
additional optimizations require extra user skills and calculation costs for tuning parameters and
hyperparameters. Thus, for example, if an ImageNet model [6, 16] that learned 1000 classes can
learn a new class "baby" from one image of a baby with neither additional training optimization nor
pretrained-DNN modification, it will be useful in actual machine learning uses.

For a DNN model trained with a sufficiently rich set of images, a reasonable representation of
unknown images must exist somewhere in the hidden multi-dimensional space. Indeed, *weight
imprinting*, proposed by Qi et al. [26], can add novel classes to Convolutional Neural Networks
(CNNs) by using final-dense-layer input of a new-class image without extra training. Qi's weight
imprinting method needs just a few CNN-architecture modifications and can provide decent accuracy
in a one-shot image classification task (e.g., accuracy for novel-class images was 21% when novel

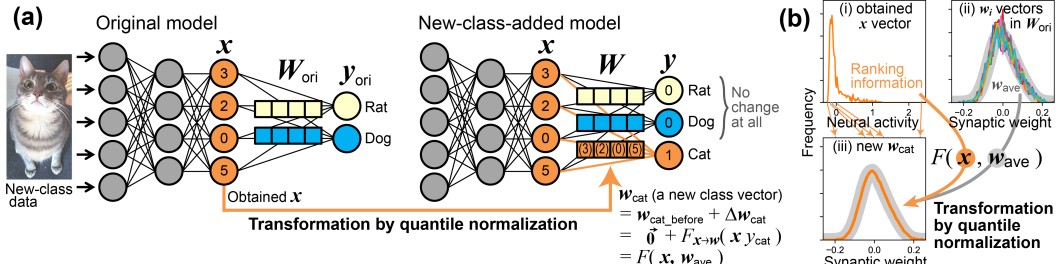

Figure 1: Scheme of DONE. (a) The neural activity input of final dense layer (orange $\boldsymbol{x}$ vector in original model) obtained from a new-class data (an image of a cat) is directly used for the transformation to the new-class vector (orange $\boldsymbol{w}_{\mathrm{cat}}$) in the new weight matrix ($\boldsymbol{W}$) without any modification to the backbone model. (b) An example case of transformation from $\boldsymbol{x}$ to $\boldsymbol{w}_{\mathrm{cat}}$, with actual distribution data when the backbone DNN is EfficientNet-B0.

100 classes were added to the original 100 classes in CUB-200-2011 dataset). Moreover, some studies [21, 36] show that the capabilities of DNN itself have the potential to enable Out-of-Distribution Detection (OOD). For the relationship between the brain and artificial computation, not only some of the features of the lower [4] and higher visual cortex [11] are explained by filtering or DNN, but also the embedding of such new-concept-learning functions into the hidden space has been analyzed [39].

In this paper, we introduce a very simple method, Direct ONE-shot learning (DONE) with a nonparametric weight imprinting. As shown in Figure 1(a), DONE directly transform the input of the final dense layer ($\boldsymbol{x}$ vector in the figure) obtained by one input image belonging to a new class (e.g., a cat in the figure) into the weight vector for the new additional class ($\boldsymbol{w}_{\mathrm{cat}}$, a row vector of the weight matrix $\boldsymbol{W}$). Then, it is done. DONE uses weight imprinting but never modifies backbone DNN including original weight matrix $\boldsymbol{W}_{\mathrm{ori}}$ unlike Qi's method. Qi's method was inspired by the context of metric learning, but DONE was inspired by Hebbian theory [2]. This difference in inspiration source makes a small but important difference in method procedures, and this study proposes a new formulation of Hebbian theory for weight imprinting.

In weight imprinting, we can assume that the new weight vector $\boldsymbol{w}_{\mathrm{cat}}$ is born out of nothing and thus is equal to its change, i.e., $\boldsymbol{w}_{\mathrm{cat}} = \vec{0} + \Delta\boldsymbol{w}_{\mathrm{cat}} = \Delta\boldsymbol{w}_{\mathrm{cat}}$. Hebbian theory is about this $\Delta\boldsymbol{w}_{\mathrm{cat}}$ and states that a synaptic weight is strengthened when both its presynaptic and postsynaptic neurons are active simultaneously. When a single image of a new class (cat) is presented as visual input, some of the presynaptic neurons $\boldsymbol{x}$ become active. Simultaneously, a postsynaptic neuron corresponding to cat is active (e.g., $y_{\mathrm{cat}} = 1$), while postsynaptic neurons for all the $i$-th original classes are not ($y_i = 0$), because the training image is known to be a cat. In a simple and conventional formulation of Hebbian theory, the change in the weight vector can be described as $\Delta\boldsymbol{w}_{\mathrm{cat}} \propto \boldsymbol{x} \cdot y_{\mathrm{cat}}$, thus $\boldsymbol{w}_{\mathrm{cat}} = \Delta\boldsymbol{w}_{\mathrm{cat}} \propto \boldsymbol{x}$, while $\Delta\boldsymbol{w}_i = 0$ because $y_i = 0$. Therefore, the mechanism of weight imprinting without modification of $\boldsymbol{W}_{\mathrm{ori}}$ can be explained by Hebbian theory.

Here, a problem arises with this simple formulation alone, because neural activity and synaptic weight are different in scale and those relationships would not be linear. For example, Figure 1(b)-(i) and (b)-(ii) show frequency distributions of neural activity in $\boldsymbol{x}$ and weight in $\boldsymbol{w}_i$, which are different in shape, in an actual DNN. If only the new $\boldsymbol{w}_{\mathrm{cat}}$ had far different statistical properties compared to the other $\boldsymbol{w}_i$, the comparison between classes would be unequal, and the additional $\boldsymbol{w}_{\mathrm{cat}}$ could inhibit the classification of the original classes (shown later). Therefore, the implementation of Hebbian theory here must include a function for the nonlinear scale transformation, i.e., $\Delta\boldsymbol{w}_{\mathrm{cat}} \propto F_{\boldsymbol{x}\to\boldsymbol{w}}(\boldsymbol{x} \cdot y_{\mathrm{cat}})$.

DONE takes into account this transformation by quantile normalization [1, 3], so that the frequency distribution of $\boldsymbol{w}_{\mathrm{cat}}$ becomes equal to that of $\boldsymbol{w}_{\mathrm{ave}}$ (the average vector of original $\boldsymbol{w}_i$ vectors), i.e., $\Delta\boldsymbol{w}_{\mathrm{cat}} = F(\boldsymbol{x}, \boldsymbol{w}_{\mathrm{ave}})$ (Figure 1(b)-(iii)). Quantile normalization is an easy and standard technique in Bioinformatics [1, 3], and it is suitable for implementing Hebbian theory. The statistical properties of $\boldsymbol{w}_{\mathrm{cat}}$ that result from the transformation from neural activity to synaptic weight should be similar to the statistical properties of original synaptic weights. For example, we could apply linear transformation

so that the mean and variance (i.e., 1st and 2nd central moments) of the elements of $w_{\text{cat}}$ are the same as those of $w_{\text{ave}}$. However, it is not clear if such adjustment for only 1st and 2nd central moments is enough in this situation where the 3rd or higher central moments could be different (such as shown in Figure 1(b)). One of the simplest solutions for every situation is to make all the statistical properties identical. As above, DONE can simply add a new class by $w_{\text{cat}} = F(x, w_{\text{ave}})$ nonparametrically, and we call this transformation *Hebbian weight imprinting* (see Methodology section for details).

Our method's basis and procedure are very simple, but it achieves similar accuracy to Qi's method and does not interfere with the original classification compared to Qi's method. DONE achieved over 50% accuracy (approximately 80% of classification of well-trained original classes) in a one-shot image classification task that adds eight new classes to a model pretrained for the ImageNet 1000 classes (ViT (Vision Transformer) [34] or EfficientNet [29]) as a backbone model (note that the chance level is less than 0.1%). In a typical five-way one-shot classification task, DONE with ViT achieved over 80% accuracy.

The advantages of DONE over other weight imprinting methods are (i) Hebbian-inspired simpler basis and procedure, (ii) no modification to backbone models, and (iii) nonparametric procedure for a wide range of backbone models including future models. The advantages of DONE as a weight imprinting are (iv) no optimization thus little calculation cost and (v) no parameters or hyperparameters thus reproducible for anyone. In addition to proposing the new methods, this paper contains the following useful information: a generic task to add new classes to 1000-class ImageNet models, and the difference in backbone DNNs, specifically, between a Transfomer (ViT) and a CNN (EfficientNet). Moreover, DONE may provide a useful insight when exploring the learning principles of the brain because DONE is inspired by the Hebbian theory.

## 2    Related work

### 2.1    One-shot and few-shot image classification

Typical learning approaches for one- or few-shot image classification are metric learning, data augmentation, and meta learning. Weight imprinting has come out from metric learning. Each of these approaches has its own advantages and purposes, and they are not contradictory but can be used in a mixed manner.

Metric learning uses a classification at a feature space as a metric space [8, 22, 30]. Roughly speaking, metric learning aims to decrease the distances between training data belonging to the same class and increase the distance between the data belonging to different classes. Metric learning such as using Siamese network [13] is useful for tasks that require one-shot learning, e.g., face recognition. A Data-augmentation approach generatively increases the number of training inputs [19, 27, 45]. This approach includes various types such as semi-supervised approaches and example generation using Generative adversarial networks [10]. Meta learning approaches train the abilities of learning systems to learn [18, 23, 43]. The purpose of meta learning is to aim to increase the learning efficiency itself, and this is a powerful approach for learning from a small amount of training data, typically one-shot learning task [41].

### 2.2    Weight imprinting

Weight imprinting is a learning method that initially arose from an innovative idea "learning without optimization" [26], and DONE is a type of weight imprinting. Weight imprinting does not contain any optimization algorithm and is basically inferior to other optimization methods by themselves in accuracy. However, comparisons of DONE with other optimization methods are useful in evaluating those optimization methods, because the performance of weight imprinting methods is uniquely determined by the backbone DNN without any randomness. Thus, weight imprinting does not aim for the highest accuracy but for practical convenience and reference role as a baseline method.

We here explain the basis of weight imprinting and then specific procedure of Qi's method. Let us consider the classification at the final dense layer of DNN models in general. In most cases, the output vector $\boldsymbol{y} = (y_1, \cdots, y_N)$ of the final dense layer denotes the degree to which an image belongs to each class and is calculated from the input vector of the final dense layer $\boldsymbol{x} = (x_1, \cdots, x_M)$, weight matrix $\boldsymbol{W}$ $(N \times M)$, and bias vector $\boldsymbol{b} = (b_1, \cdots, b_N)$. Here, for $i$-th class in $N$ classes $(i = 1, 2, \cdots, N)$, a scalar $y_i$ is calculated from the corresponding weight vector $\boldsymbol{w}_i = (w_{i1}, \cdots, w_{iM})$ ($i$-th row vector of $\boldsymbol{W}$ matrix) and bias scalar $b_i$ as the following equation:

$$y_i = \boldsymbol{x} \cdot \boldsymbol{w}_i + b_i = ||\boldsymbol{x}||_2 \, ||\boldsymbol{w}_i||_2 \cos \theta + b_i, \tag{1}$$

where the cosine similarity is a metric that represents how similar the two vectors $\boldsymbol{x}$ and $\boldsymbol{w}_i$ are irrespective of their size. Thus, this type of model contains cosine similarity as a part of its objective function. It is also possible to use the cosine similarity alone instead of the dot product [25].

Weight imprinting uses this basis of the cosine similarity. The cosine similarity will have the maximum value 1 if $\boldsymbol{x}$ and $\boldsymbol{w}_i$ are directly proportional. Thus, if a certain $\boldsymbol{x}$ is directly used for the weight of a new $j$-th class $\boldsymbol{w}_j$ $(j = N + 1, \cdots)$, the cosine similarity for $j$-th class becomes large when another $\boldsymbol{x}$ with a similar value comes.

In Qi's method, to focus only on the cosine similarity as a metric for the objective function, the backbone DNN models are modified in the following three parts:

- **Modification 1** : Adding $L_2$ normalization layer before the final dense layer so that $\boldsymbol{x}$ becomes unit vector, i.e., $||\boldsymbol{x}||_2 = 1$

- **Modification 2** : Normalizing all $\boldsymbol{w}_i$ to become unit vectors, i.e., $||\boldsymbol{w}_i||_2 = 1$ for all $i$.

- **Modification 3** : Ignoring all bias values $b_i$, i.e., $\boldsymbol{b}$ vector.

Then, the final-dense-layer input obtained from a new-class image $\boldsymbol{x}_{\text{new}}$ ($L_2$-normalized, in Qi's method) is used as the weight vector for the new class $\boldsymbol{w}_j$, i.e.,

$$\boldsymbol{w}_j = \boldsymbol{x}_{\text{new}}. \tag{2}$$

Qi's method is already simple and elegant, but it still requires some modifications to the backbone DNN, which involves changes in the objective function. Whether a modification is good or bad depends on the situation, but if not necessary, it would be better without modification in order to avoid unnecessary complications and unexpected interference with the original classification because the backbone DNN would be already well optimized for a certain function. Also, Qi's method uses linear transformation for conversion of $\boldsymbol{x}$ into $\boldsymbol{w}_j$ as a result of focusing on the cosine similarity, without considering the difference in statistical properties between $\boldsymbol{x}$ and $\boldsymbol{w}_j$, which limits the range of backbone DNNs used. There have been various researches that make Qi's method more complex and applicable [32, 38, 40, 44, 46], but to the best of our knowledge, none that make it simpler or solve the transformation problem.

## 3 Methodology

### 3.1 Procedure and basis of DONE

DONE does not modify backbone DNN and just directly applies $\boldsymbol{x}_{\text{new}}$ to $\boldsymbol{w}_j$ $(j = N + 1, \cdots)$, as shown in Figure 1, as

$$\boldsymbol{w}_j = F(\boldsymbol{x}_{\text{new}}, \boldsymbol{w}_{\text{ave}}), \tag{3}$$
$$b_j = \tilde{\boldsymbol{b}}_{\text{ori}}, \tag{4}$$

where $F(\boldsymbol{x}_{\text{new}}, \boldsymbol{w}_{\text{ave}})$ is a quantile normalization of $\boldsymbol{x}_{\text{new}}$, using the information of the average weight vector for original classes ($\boldsymbol{w}_{\text{ave}}$) as the reference distribution, and $\tilde{\boldsymbol{b}}_{\text{ori}}$ is the median of the original bias vector $\boldsymbol{b}_{\text{ori}}$. Then, it is done.

In the quantile normalization, the elements value of resultant $w_j$ become the same as that of the reference $w_{ave}$. Specifically, for example, first we change the value of the most (1st) active neuron in $x_{new}$ vector into the highest (1st) weight value in $w_{ave}$. We then apply the same procedure for the 2nd, 3rd, $\cdots$, $M$-th highest neurons. Then, the ranking of each neuron in $x_{new}$ remains the same, and the value of each ranking is all identical to $w_{ave}$. This resultant vector is $w_j$. Namely, all statistical properties of the elements of $w_j$ and $w_{ave}$ are identical (the frequency distributions are the same).

For $w_{ave}$, to implement the concept of Hebbian theory, we used all the $N \times M$ elements of flattened $W_{ori}$, divided these elements into $M$ parts in ranking order, and obtained the $M$ median values in each $N$ element as $M$-dimensional $w_{ave}$. For example, in the case of ViT-B/32 ($N = 1000$, $M = 768$), the highest value of $w_{ave}$ is the median of 1st to 1000th highest elements in 768,000 elements of $W_{ori}$, and the lowest value is the median of 767,001th to 768,000th elements.

## 3.2 Limitations, applications, and potential negative societal impacts

As limitations, DONE requires a neural network model that has dense layer for classification as above. DONE can be used for a wide range of applications with DNN classifiers, including OOD applications[42]. There can be various potential negative societal impacts associated with these broad applications. One example is immoral classification or discrimination when classifying human-related data, such as facial images, voices, and personal feature data.

## 3.3 Implementation and Dataset

As backbone models, we employed ViT-B/32 [34] and EfficientNet-B0 [29] as two representative DNNs with different characteristics, using "vit-keras" [48] and "EfficientNet Keras (and TensorFlow Keras)" [47], respectively. Also, for transfer learning, we employed InceptionV1 [15] (employed in the paper by Qi et al. [26]) and ResNet-12 [20], using "Trained image classification models for Keras" [49] and Tensorflow [12], respectively. All models used in this study were pretrained with ImageNet.

We used CIFAR-100 and CIFAR-10 [7] for additional classes, using Tensorflow [12]. Also, for transfer learning, we used CIFAR-FS [28] by Torchmeta [31]. We used ImageNet (ILSVRC2012) images [6, 16] for testing the performance of the models. We used information of 67 categorization [33] of ImageNet 1000 classes, for a coarse 10 categorization in Figure 4(a). All images were resized to $(224 \times 224)$ by OpenCV [50] or the preprocessing resizing layer of Tensorflow [12].

# 4 Results and Discussion

## 4.1 One-class addition by one-shot learning

First, according to our motivation, we investigated the performance of DONE when a new class from one image was added to a DNN model pretrained with ImageNet (1000 classes). As new additional classes, we chose eight classes, "baby", "woman", "man", "caterpillar", "cloud", "forest", "maple_tree", and "sunflower" from CIFAR-100, which were not in ImageNet (shown in Figure 2(a)). The weight parameters for the additional one class $w_j$ is generated from one image, thus the model had 1001 classes. To conduct stochastic evaluations, 100 different models were built by using 100 different training images for each additional class.

Figure 2(b) shows letter-value plots of the accuracy for each additional class (chance level $1/1001$). The mean of the median top-1 accuracy of 8 classes by DONE were 56.5% and 92.1% for ViT and EfficinetNet, respectively (black line). When the mean accuracy was compared with the accuracy of ImageNet validation test by the original 1000-class model (orange line; 65% and 69%), the mean with ViT had no significant difference and the mean of EfficinetNet was significantly greater (one sample $t$-test; two-sided with $\alpha$=0.05, in all statistical tests in this study). The higher accuracy than the original classes in EfficientNet is strange, and it is considered that EfficientNet tends to recognize the new-class images as just OOD (see later, Figure 4).

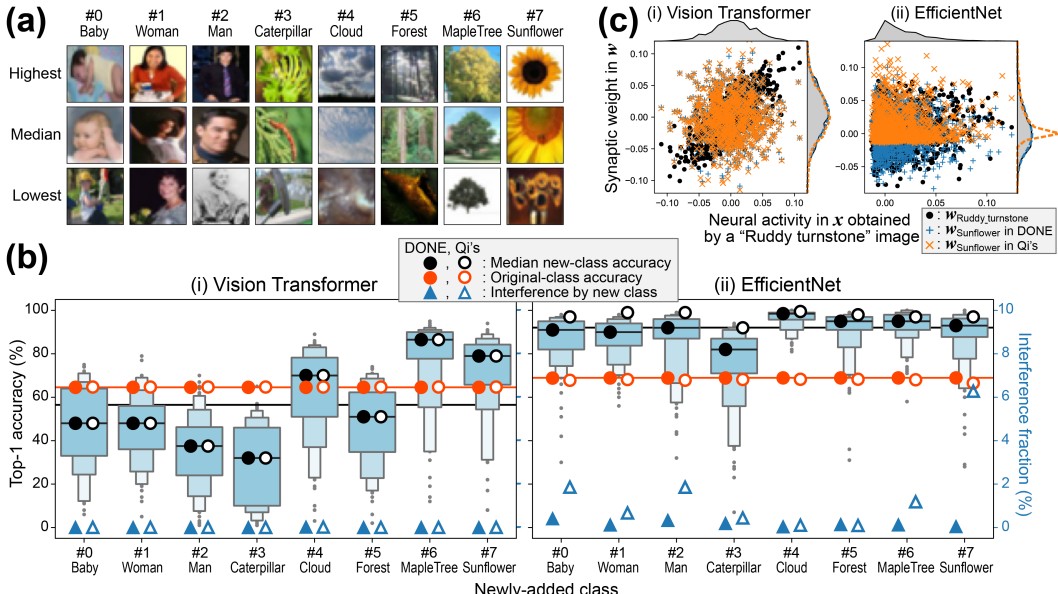

Figure 2: One-class addition by one-shot learning. (a) Representative images of the newly-added CIFAR-100 classes. Each image was chosen as a representative because the model that learned the image showed the highest, median, and lowest accuracy in each class in (b)-(i). (b) Letter-value plots of top-1 accuracy of the one-class-added models obtained by one-shot learning with DONE in classification of new-class images. The median top-1 accuracy of the new-class classification (black circles), top-1 accuracy in original-class classification (orange circles), and the fraction of the interference with the original-class classification by the newly-added class (blue) are also plotted for DONE (closed) and Qi's method (open). Black and orange lines show the mean of the 8 closed circles. (c) The relationship between $x$ and $w$ vectors when an image of "Ruddy turnstone" is input and it is miss-classified as "Sunflower" only in the case of Qi's method with EfficientNet. The frequency distributions of elements of each vector are also shown outside the plot frames.

An obvious fact in one-shot learning is that a bad training image produces a bad performance, e.g., the accuracy was 6% in ViT when the training image was a baby image shown at the bottom left in Figure 2(a). But in practical usage, a user is supposed to use a representative image for the training. We therefore think that the low performance due to a bad training image is not a significant issue.

We investigated the interference of the class addition with the classification performance of the original 1000 classes. We evaluated the original 1000-class model and eight 1001-class models that showed the median accuracy, by using all 50,000 ImageNet validation images (Figure 2(b)). The difference between the accuracy of the original 1000-class model (orange line) and the mean accuracy of the eight 1001-class models (orange closed circles) was less than 1% (0.004% and 0.664% for ViT and EfficinetNet, respectively).

Figure 2(b) also shows the fraction of ImageNet validation images in which the output top-1 answer of the added model was the new class (thus incorrect) in the 50,000 images (blue closed triangles; right axis). This interference fraction was low in ViT, and for example, only 2 images out of 50,000 were classified as "baby". When we checked the two images, both images indeed contained a baby though its class in ImageNet was "Bathtub". Therefore, observed interference in ViT was not a mistake but just the result of another classification. EfficientNet shows a significantly greater fraction of interference than ViT (Wilcoxon signed-rank test), but we also confirmed that a similar thing happened, e.g., 198 of the 204 ImageNet-validation images classified as "baby" in EfficientNet contained human or doll.

We also compared DONE with Qi's method. Open circles and triangles show the results using Qi's method instead of DONE in the same tests described above. When the backbone model

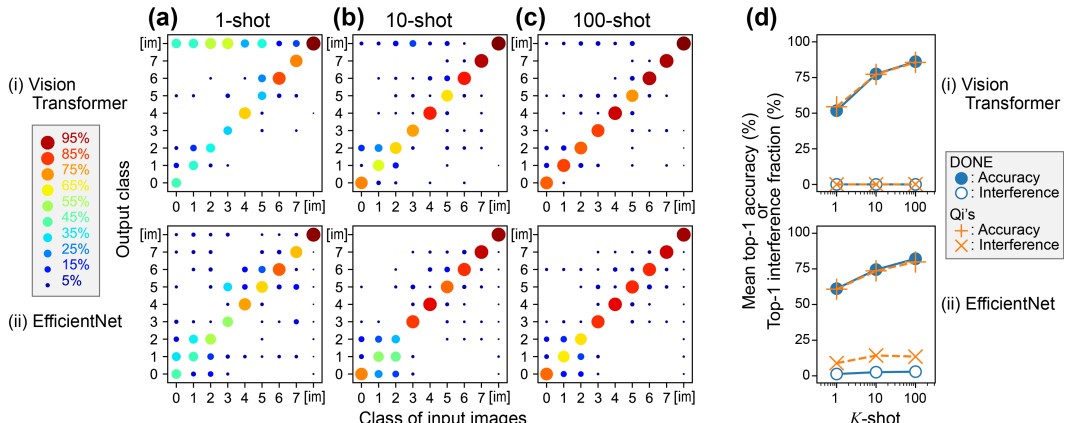

Figure 3: Multi-class addition and $K$-shot learning. (a), (b), and (c) show the results of the 1008-class model constructed by 1, 10, and 100-shot learning, respectively. The horizontal and vertical axes show the class of the input images, and the output class, respectively. The class numbers are those shown in Figure 2(a). The class [im] contains 1000 classes of ImageNet. (d) Summary of the mean accuracy and the interference fraction with original-class classification by DONE and Qi's method.

was EfficientNet, the strangely-high accuracy (paired sample $t$-test) and the interference fraction (Wilcoxon signed-rank test) were significantly greater by Qi's method than by DONE. Also, a significant outlier of decreased accuracy in the ImageNet validation test was observed (orange open circle for "Sunflower"; Smirnov-Grubbs test). On the other hand, those differences were not significant in the case of ViT.

To investigate the cause of the difference between DONE and Qi's method, especially about the greater interference by Qi's method in EfficientNet, we plotted $w_{\text{Sunflower}}$ and $w_{\text{Ruddy\_turnstone}}$ against $x$ obtained from an image of "Ruddy turnstone" (Figure 2(c)). Note that all the vectors here are $L_2$-normalized, and thus DONE and Qi's method have common $w_{\text{Ruddy\_turnstone}}$ and $x$. In the case of ViT, the shape of the frequency distributions of all these vectors are similar, and $w_{\text{Sunflower}}$ of DONE and Qi's method are similar. On the other hand, in EfficientNet, the shape of frequency distributions are more different between $w_{\text{Ruddy\_turnstone}}$ and $x$ than ViT, and thus the shape of frequency distributions are more different between $w_{\text{Ruddy\_turnstone}}$ and $w_{\text{Sunflower}}$ by Qi's method than by DONE. Then, by Qi's method, $x$ is more similar to $w_{\text{Sunflower}}$ than $w_{\text{Ruddy\_turnstone}}$ because not neuronal match but statistical properties are similar. This is the basis of the problem by a linear transformation of neural activity to synaptic weight. Therefore, the difference between DONE and Qi's method appears in the interference when the statistical properties of $x$ and $w_i$ vectors in the backbone DNN are different (thus the results in ViT are similar between DONE and Qi's method).

## 4.2 Multi-class addition and K-shot learning

DONE was able to add a new class as above, but it might just be because the models recognized the new-class images as OOD, i.e., something else. Therefore, it is necessary to add multiple new similar classes and check the classification among them. In addition, it is necessary to confirm whether the accuracy increase by increasing the number of training images, because in practical uses, users will prepare not just one training data but multiple training data for each class.

Specifically, we used one image from each of the eight classes and added new eight classes to the original 1000 classes, using DONE as one-shot learning. We evaluated this 1008-class model by 100 CIFAR test images for each of 8 classes and 10,000 ImageNet validation images. Figure 3(a) shows the results of the output of the representative model constructed by one-shot learning in which one image that showed median accuracy in Figure 2(b) was used as a standard training image of each class. In both backbone DNNs, the fraction of output of the correct class was the highest among the 1008 classes, and mean top-1 accuracy of the 8 classes was 51.8% and 61.1% in ViT and EfficientNet,

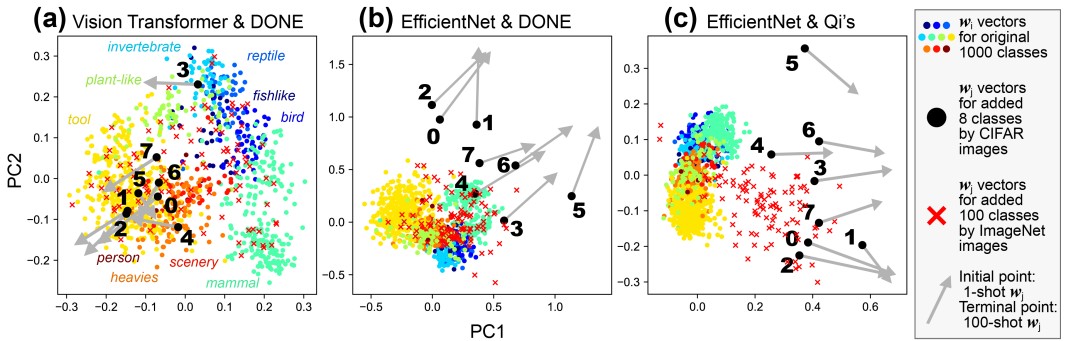

Figure 4: Principal component analysis of weight vectors. PCA of each $w_i$ and $w_j$ vector in the one-shot 1008-class models shown in Figure 3(a). Different colors for $w_i$ show a coarse 10 categorization of the classes. Also, 100 $w_j$ vectors obtained by inputting 100 ImageNet images are shown.

respectively. That is, DONE was also able to classify newly added similar classes together with the original classes, in both DNNs.

Next, we increased the number of training images as $K$-shot learning. In the case of 10-shot learning (Figure 3(b)), each of the ten images was input to obtain each $x$, and the mean vector of the ten $x$ vectors was converted into $w_j$, according to the Qi's method. For this representative 10-shot model, we used 10 images whose index in CIFAR-100 was from the front to the 10th in each class. We also tested 100-shot learning in the same way (Figure 3(c)). As a result, we found that such a simple averaging operation steadily improved the accuracy (Figure 3(d) summaries the mean accuracy).

When we used Qi's method, compared to the case of DONE, ImageNet images were significantly more often categorized to the new classes as interference only when the backbone model was EfficientNet (paired sample $t$-test), while there was no significant difference in the mean accuracy of 1, 10, 100-shot 1008-class models for the added 8 classes between DONE and Qi's method with both backbone DNNs (Figure 3(d)). Thus, again the interference in the case of EfficientNet is significantly greater in Qi's method than DONE.

### 4.3 Principal component analysis of weight vectors

Qi's method showed greater interference in classification of the original-class images than DONE only when the backbone DNN is EfficientNet. Moreover, even by DONE, EfficientNet showed greater interference than ViT and strangely-high accuracy at 1001-class model, even though DONE did not change the weights for the original classes and transformed the new-class weights so that the statistical properties were the same as those of the original-class weights. Therefore, there should be at least two reasons for these results only shown in EfficientNet, and DONE cannot correct at least one of them. To investigate those reasons, we analyzed $W$ matrix ($w_i$ and $w_j$ vectors) of the one-shot 1008-class models shown in Figure 3(a) (and corresponding models by Qi's method) by Principal component analysis (PCA; Figure 4).

In ViT by DONE (Figure 4(a); Qi's methods showed similar results, see Figure S1), newly added $w_j$ vectors (black circles, with the ID number of newly-added 8 classes) were comparable to those of the original classes $w_i$ (colored circles), e.g., $w_j$ vector of a new class "caterpillar (3 in Figure 4(a))" was near $w_i$ of original "invertebrate" classes. Also, even when we got $w_j$ by inputting ImageNet images (red crosses; validation ID from the front to the 100th), those ImageNet $w_j$ vectors distributed in similar range.

On the other hand, in EfficientNet by DONE (Figure 4(b)), most of newly-added 8-class $w_j$ were out of the distribution (meaning out of minimal bounding ellipsoid) of $w_i$ of original 1000 classes, while most of the ImageNet $w_j$ (red crosses) were inside the distribution of $w_i$. Therefore, in the case of DONE, the main reason for the observed greater interference and strangely-high accuracy

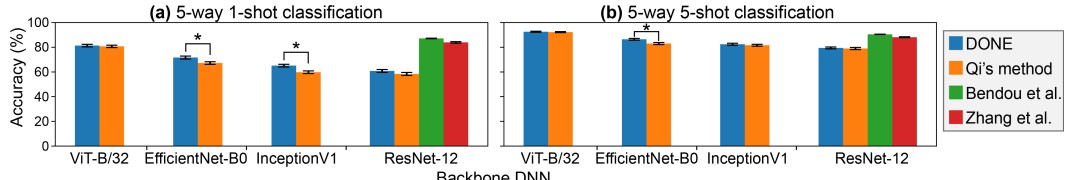

Figure 5: 5-way 1-shot (a) and 5-shot (b) classification accuracy on CIFAR-FS with various backbone DNNs. Error bars show standard errors. Asterisks mean significant differences between DONE and Qi's method (Dwass-Steel-Critchlow-Fligner test).

in EfficientNet than ViT would be the difference between ImageNet and CIFAR. These results are consistent with known facts that ViT is considered to be better at predictive uncertainty estimation [24, 37], more robust to input perturbations [35], and more suitable at classifying OODs [36] than CNNs like EfficientNet.

In EfficientNet by Qi's method (Figure 4(c)), most of not only 8-class $w_j$ but also the ImageNet $w_j$ were out of the distribution of $w_i$ of original 1000 classes. The difference in the distributions between the original $w_i$ and the ImageNet $w_j$ is considered to indicate the difference in the statistical properties of $x$ and $w_i$ vectors in EfficientNet.

In the case of 100-shot learning (the terminal points of the gray arrows in Figure 4), $w_j$ went away from the cluster of original $w_i$ in all three cases, although their performance was better than one-shot learning. Therefore, 100-shot $w_j$ were considered to work somehow in a different way from the original $w_i$.

## 4.4 Transfer few-shot learning

DONE is recommended for the easy addition of new classes, not for transfer learning. However, DONE can work for it (Figure S2) and is convenient for the evaluation of DNNs and other few-shot learning methods. We examined the 5-way (5 classes) 1-shot task of CIFAR-FS, which is a kind of standard task in one-shot classification. Specifically, we used each single image in 5 classes out of 100 classes of CIFAR-100 for constructing a model, and evaluate the model by 15 images in each class. The combination of the 5 classes (and corresponding training images) was randomly changed in 100 times (Figure 5(a)). Also 5-way 5-shot task was tested in a similar way (Figure 5(b)).

We found ViT significantly outperformed the other DNNs in all conditions (Dwass-Steel-Critchlow-Fligner test) by both DONE and Qi's method. Compared to Qi's method, DONE shows significantly greater accuracy with some CNN models, and never significantly worse, although it is not an expected advantage of DONE and there would be no particular reason for it.

Figure 5 also clearly shows that how much other state-of-the-art one-shot learning methods with optimization (methods in [43] and [45]) outperform DONE as the baseline without optimization, at the same test with a common backbone DNN (ResNet-12).

## 5 Conclusion and Future work

This paper has proposed DONE, one of the simplest one-shot learning methods that allows us to add new classes to a pretrained DNN at a decent accuracy without optimization or modification of the DNN. DONE applies Hebbian weight imprinting, which is a new implementation of Hebbian theory by quantile normalization, to the final dense layer of a DNN model. Given the simplicity and wide applicability, not only DONE but also Hebbian weight imprinting alone are expected to be applied in a wide range of the field of neural networks. This study has just proposed the method, and its scalability (Figure S3) and expected applications are yet to be elucidated. Since the performance of DONE is completely dependent on backbone DNNs and further development of DNN is certain, the situation to obtain sufficient accuracy with DONE may soon come.

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

# A Appendix

Code and data are submitted as supplemental material (the same as 1st submission). Supplemental figures are added in this revised submission (NeurIPS_FigureS.pdf).

