# OpenReview forum: "It's DONE: Direct ONE-shot learning with Hebbian weight imprinting"
_NeurIPS.cc/2022/Conference — NeurIPS 2022 Submitted_

### Official Review · Reviewer_8B2d · 2022-07-10

**Rating:** 5
**Confidence:** 3
**Soundness:** 3 good
**Presentation:** 2 fair
**Contribution:** 3 good

**Summary:**

The paper presents a method for the one-shot classification of novel inputs. For the classification of novel inputs, the method uses representations obtained by the final dense layer of a pretrained backbone model to create novel weight vectors of new classes. Importantly, DONE does not optimize or change the backbone model in any way, making it applicable to various backbone models and computationally efficient.

**Questions:**

My main suggestion is that you would make a further effort to convince this method's benefits over Qi's. In the current paper, the benefits are argued in text. Try to argue using a test where Qi's modification to the backbone model impairs performance, while DONE does not.

Also, I suggest adding more details about the practical implementation of quantile normalization.

Finally, the paper needs to be thoroughly edited.

**Limitations:**

The paper adequately discusses the method limitations.

**Strengths And Weaknesses:**

Strengths:
1) Introducing quantile normalization in machine learning, a method that was unknown to me.

Weaknesses:
1) The link to Hebbian learning is weak and not meaningful for this paper goals.
2) The idea that the last dense layer of a DNN includes representations that allow novel classifications is not novel, without any model modification, is not novel (e.g., [1] analyzed it theoretically).
3) There are many writing problems. Too many to mention them all. Examples: a. talking about the heritage of mankind in line 135 (unprofessional); b. figure 2 has an irrelevant screen cursor at the right-most data point. More importantly, essential legend details of this figure appear in the text instead of the caption, which is confusing; c. “the simplest” in line 324 is also not professional (I think that Qi’s method is simpler).
4) The method does not improve upon previous works Figure 2 shows that Qi’s method works similarly or better. I am not convinced that the backbone modifications it does are detrimental.
5) Accepting the updated paper that was submitted via the supplementary materials, at a later deadline, is unfair to other authors.

[1] Sorscher, B., Ganguli, S., & Sompolinsky, H. (2021). The geometry of concept learning. bioRxiv.‏

---

> ### Author Response · Authors · 2022-08-02
> **Thank you for your comments and essential ideas for improving our manuscript.**
>
>
> Thank you for your comments and essential ideas for improving our manuscript. Now we believe the value of our paper has remarkably increased.
>
> We really appreciate your idea, in particular, the idea of presenting a comparison with the previous method in figures rather than text is essential for the improvement.
>
> We thoroughly edited our manuscript according to your comments and ideas as described below.
>
> > Strengths:
> > Introducing quantile normalization in machine learning, a method that was unknown to me.
>
> Thank you for your understanding. Quantile normalization actually has been used in machine learning field (very rare though; e.g., Yang and Shami, arXiv.2201.11812.), but has never been used for implementation of Hebbian learning. Because of your and other reviewer's encouraging comments, we decided to emphasize quantile normalization and its application to Hebbian implementation.
>
>
> > The link to Hebbian learning is weak and not meaningful for this paper goals.
>
> We think this comment is due to our poor description. Our paper provides a new math formation for Hebbian theory and shows that it works. Thanks to the reviewers, the description for the link to Hebbian learning in the revised manuscript is dramatically improved. We believe that the revised manuscript conveys that Hebbian learning is meaningful both for the goals of the paper and researchers related to NeurIPS.
>
> > The idea that the last dense layer of a DNN includes representations that allow novel classifications is not novel, without any model modification, is not novel... (e.g., [1] analyzed it theoretically).
>
> We agree that those alone are not novel, especially in the fields of metric learning or out-of-distribution detection. Weight imprinting is one of the typical examples to apply it for class addition task. We have included the paper [1] in the part of these explanations in “Introduction” section (line 38). Thank you for giving us a very interesting paper.
>
> > There are many writing problems. ...
>
> Thank you for your advice. We thoroughly edited our manuscript. For example, the arrow is not the screen cursor, but we explained it in the text. We agree that such misinterpretations are all due to our poor description. As you pointed out, these problems do not occur if the explanation is in the caption instead of in the text.
>
>
> > The method does not improve upon previous works Figure 2 shows that Qi’s method works similarly or better. I am not convinced that the backbone modifications it does are detrimental.
>
> Our method overcomes a problem with Qi's method, but our writing was bad. To clarify the difference between our method and Qi's method, we have made a major improvement to include the comparison results in all figures except scheme (i.e., Figures 2-5), which is your idea as below.
>
> The higher accuracy by Qi's method with EfficientNet in Figure 2 does not mean high performance. This is because Qi's method with EfficientNet just tends to respond to any input as a new class. Thus, it is not an advantage but a problem. We added this explanation (line 227).
>
> We do not think that backbone modifications are detrimental to functionality. We just think it is better not to modify it if it is not necessary/effective (and added a sentence: line 140). We think it is common in modeling.
>
> In any case, all of these are considered to be misunderstanding due to our bad descriptions, which have been corrected in the revised manuscript.
>
>
> > Accepting the updated paper that was submitted via the supplementary materials, at a later deadline, is unfair to other authors.
>
> We agree with it. We found a coding error when we were preparing the code for the supplemental material, and we thought it would be fair to let reviewers know it as soon as possible, although those changes did not affect the claims of the paper.
>
> > My main suggestion is that you would make a further effort to convince this method's benefits over Qi's...
>
> We strongly agree with this comment and idea, and we have revised our paper with this idea. As described above, now all figures (except the scheme) include comparison results with Qi's method. We also added figures to explain why (Figure 1b and 2c). We feel those revisions have dramatically improved the quality of the paper. Thank you for the important and essential idea.
>
>
> > Also, I suggest adding more details about the practical implementation of quantile normalization.
>
> Thank you for the suggestion. We agree with it and have added the details (line 156).
>
> > Finally, the paper needs to be thoroughly edited.
>
>
> Thank you for allowing us to make major revisions. We indeed thoroughly edited our manuscript according to comments and ideas of you and the other reviewers. We feel that this revision has greatly improved the value of our paper.
>
> Finally, we are very grateful for your time and essential ideas for improving our manuscript, which absolutely increased the value of our paper.

---

> > ### Comment · Reviewer_8B2d · 2022-08-05
> > **Reviewer response**
> >
> > Thank you for addressing my comments. Your updated manuscript is indeed much better than the previous one.
> >
> > I changed my rating to 5, borderline accept. The paper presents a simple method for defining new classes based on their neural activity in the last layer of a pretrained network. It is a nice small addition to the literature. However, there are still some flaws. Mainly, the relation the Hebbian learning is irrelevant. The results do not teach anything new about Hebbian learning or the brain. Therefore, the paper's relation to Hebbian learning in that Hebbian learning is an $\textit{interpretation}$ of the model. As such, its mention belongs to the Discussion. Another flaw is that the evaluation is limited to only two backbones and one image classification task. As the two backbones show different results, further comparison with other models would have been beneficial.

---

> > > ### Author Response · Authors · 2022-08-05
> > > **Thank you very much for reading and commenting on our revised manuscript.**
> > >
> > > Thank you very much for reading and commenting on our revised manuscript.
> > >
> > > We are grateful that you, as a reviewer who commented on the essential points, gave us a higher evaluation.
> > >
> > > In addition, we would like to thank you again for raising the following important discussion points.
> > >
> > > > However, there are still some flaws. Mainly, the relation the Hebbian learning is irrelevant. The results do not teach anything new about Hebbian learning or the brain. Therefore, the paper's relation to Hebbian learning in that Hebbian learning is an interpretation of the model. As such, its mention belongs to the Discussion.
> > >
> > > Thank you for your important comments regarding the paper's relation to Hebbian learning. We agree that it is an "interpretation" and its mention belongs to the Discussion.
> > >
> > > This paper presents the model's relation to Hebbian learning and the performance of the model. In hindsight (i.e., now), " Hebbian-inspired model was useful" can be the same as "the model built for performance was related to Hebbian." Therefore, we think it can be both a "motive" and an "interpretation."
> > >
> > > We think both "the mention belongs to the Introduction" and "the mention belongs to the Discussion" have pros and cons. We think that the easiest way for readers to understand why we introduced Weight imprinting and Quantile normalization among many methods is that the mention belongs to the Introduction.
> > >
> > > Therefore, we think that the mention is better in the Introduction, but we believe the discussion here is so constructive that we would like to reflect it in the paper, e.g., it might be worth stating like "We here explain the interpretation of the relationship between our method and Hebbian theory" at the part in the Introduction. Also, we think it might be better to add a sentence like "In this paper, Hebbian theory was described as a source of inspiration and an interpretation of the model, but it is also expected to deepen our understanding of the brain and Hebbian theory through further analysis using various backbones and tasks, as well as neuroscientific experiments” in the “Conclusion and Future work.”
> > >
> > >
> > > > The results do not teach anything new about Hebbian learning or the brain.
> > >
> > > We believe that our interpretation is also useful for understanding the brain (and we feel you also probably mean similar things by "interpretation"). For example, suppose someone has created a model that can explain/reproduce an unexplained phenomenon/function. Its explanation/reproduction does not imply that the cause of the phenomenon/function is the same as the principle of the model. However, it amounts to proposing a hypothesis that was not rejected in at least one aspect. Therefore, by conducting next research (e.g., brain experiments) according to that hypothesis, we can make further challenges toward understanding the phenomenon/function.
> > >
> > > This paper is also not useless for understanding the brain. For example, we can study the brain by considering what the new-y addition here is in the brain (ycat, in Figure 1). One might hypothesize that it is a top-down function of the brain. It may allow us to ask about higher-order functions such as the frontal cortex. We might be able to investigate that with fMRI in tasks where humans are learning new concepts.
> > >
> > >
> > > > Another flaw is that the evaluation is limited to only two backbones and one image classification task. As the two backbones show different results, further comparison with other models would have been beneficial.
> > >
> > > We agree with this comment and will add results with other backbones in the supplemental material. In all the CNNs we used, the frequency distributions of x vectors were right-tailed and the distributions of w vectors were bell-shaped. In other words, only ViT showed bell-shaped x vectors.
> > >
> > > Although not directly related to this paper, we here discuss the x distribution in the brain, relating to the above discussion on understanding the brain. At first, we thought that the right-tailed would be a more suitable distribution of neural firings. However, after considering the possibility that it is bell-shaped, we thought x could be considered as activity of neural clusters instead of neurons. This would be consistent with previous macaque experimental results suggesting that an object was represented by a combination of multiple neural clusters, each representing a visual feature [Tshunoda 2001]. Therefore, it would not be strange for the frequency distribution of x to be bell-shaped like ViT. In this way, it might be interesting to consider the distribution of x in the brain (although we do not argue it would be directly useful).
> > >
> > > K. Tsunoda, Y. Yamane, M. Nishizaki, and M. Tanifuji, “Complex objects are represented in macaque inferotemporal cortex by the combination of feature columns,” Nat Neurosci, vol. 4, no. 8, Art. no. 8 , Aug. 2001, doi: 10.1038/90547.
> > >
> > > We sincerely appreciate your time.

---

### Official Review · Reviewer_j4P6 · 2022-07-11

**Rating:** 5
**Confidence:** 4
**Soundness:** 1 poor
**Presentation:** 2 fair
**Contribution:** 2 fair

**Summary:**

The paper proposes a one-shot learning mechanism that adds a new class to the network's output using quantile normalization of the new input (to the last layer) w.r.t. weights in the last layer that correspond to the old classes.

**Questions:**

## Large concerns
### Hebbian learning
This work does not describe a Hebbian mechanism of learning. Hebbian means $\Delta W_{ij} = f(x_i, y_j)$ for input neuron $x_i$ and output neuron $y_j$ (in the classical Hebbian sense, $f(x, y)=xy$). Eq. 3 is fundamentally not Hebbian because it uses information about other weights. Lines 154-155 say quantile normalization "is suitable for implementing Hebbian theory", but this claim is not backed up. The authors should either thoroughly explain what makes their method Hebbian, or remove mentions of Hebbian learning.

### Comparison with previous methods
In lines 104-109 the authors try to explain why they’re not comparing their method to previous work directly:
> It is meaningless to compare the above three approaches with weight imprinting, because weight imprinting does not contain any optimization algorithm. Therefore, in principle, there is no reason for weight imprinting methods to outperform other methods by themselves in accuracy. The performance of weight imprinting methods is uniquely determined by the backbone DNN without any randomness, hence its performance is suitable as a reference baseline for other methods. Thus, weight imprinting does not aim for the highest accuracy but for practical convenience and reference role as a baseline method.

– I can agree with not chasing the best accuracy, but then there must be a discussion of concrete practical benefits of the method. (No optimization != faster, as quantization takes some time too.)

### Overall reason behind the method
Section 3.1: I did not understand why quantile normalization helps. Shouldn’t it make the response in each neuron $y$ similar to that of the new class? Either way, the authors should spend more time explaining their method and why it works, as it is the core contribution.

## Unclear parts in the text

Lines 135-136: "the backbone DNN model is a very good model as a heritage of mankind and should not be changed as much as possible (especially for many non-expert users)"
– this is not an explanation.

153: what probability distributions? Everything has been deterministic so far, so it's worth explaining what distributions are discussed here. Is it over individual weights? Individual neurons?

166: "The range of applicable models is yet unclear, but in principle it is wider than Qi’s method."
– what does it mean? The line before said that you need a network with a dense final layer, which clearly defines the range of models.

223-224: what’s good accuracy for practical uses? It’s never explained how the authors came up with that value

## Small corrections
**Note**: the paper contains a lot of grammatical errors. One simple way to fix it is to copy-paste the source code into a google doc and go through all suggested corrections.

Line 15 (abstract): performs "at a", not "a", practical-level accuracy?

Also next sentence: Can write as "DONE overcomes ...mentioned issues of DNNs..."; Currently the sentence doesn’t read well.

92: uses "a" classification

93: aim"s"

145: applies to, not apply into

212-214: black and orange said twice

Why is the supplementary material just a corrected version of the main paper?


**Limitations:**

The authors can mostly addressed the limitations of their work (although see below), but they did not account for the potential negative impacts. I'm not flagging it for an ethics review as it's just a small correction though.

Checklist 1C: the described model can be easily adapted for face recognition. This does imply potential negative societal impacts and should be discussed by the authors, even though previous work can be used in a similar way.

Related, the authors refer to practical uses throughout the whole paper, in particular when talking about performance of DONE. But it’s never discussed what those are, and why the achieved performance of the method is good enough for them.

**Strengths And Weaknesses:**

## Strengths:
1. The proposed one-shot classification method is simple and, unlike previous work, doesn't require layer normalization in the second last layer.
2. The method works reasonably well.
3. The method is novel, as far as I know.

## Weaknesses:
1. This work has "Hebbian" in the title and some part of the text, but as far as I can tell has nothing to do with Hebbian learning. (Elaborated in the Questions section)
2. Performance of the method is not adequately compared to previous work. (Again, elaborated below)

## Summary

My current score is 4 (borderline reject). The paper proposed an interesting method, but it doesn't properly explain why it works, and it doesn't justify the lack of comparison (in terms of performance and computational complexity) with previous methods. I'm willing to increase the rating if my points are addressed, however.

---

> ### Author Response · Authors · 2022-08-02
> **Thank you for your to-the-point comments and great ideas for explaining the core part.**
>
>
> Thank you for your to-the-point comments and great ideas for explaining the core part, which taught us not just what to revise but also how to revise the manuscript.
>
> We really appreciate it, in particular, you gave us a concrete idea of how we should explain the relationship with Hebbian theory, as well as many other necessary comments for improvement.
>
> Below are our answers to all your comments in "weaknesses" and "questions".
>
> > This work has "Hebbian" in the title and some part of the text, but ...
>
> Our paper provides a new mathematical formation for Hebbian theory and shows that it works, but our writing was bad. As described below, the relationship with Hebbian theory is clearly explained in the revised manuscript, and we feel that the paper has improved dramatically, thanks to your idea.
>
> > Performance of the method is not adequately compared to previous work...
>
> We agree with this comment. We have made a major improvement to include comparison results with the previous method in all figures except scheme (i.e., Figures 2-5), as described below.
>
> > My current score is 4 (borderline reject). The paper proposed an interesting method, ...
>
> Thank you for your precise and encouraging comments. According to your comments, we revised the entire manuscript, as described below.
>
> > Hebbian learning
> > This work does not describe a Hebbian mechanism of learning. Hebbian means dWij=f(xi,yj) ...
>
> This comment was very informative for us, and we learned a lot from it. There is certainly a strong relationship with Hebbian learning, but we once considered removing the mentions of Hebbian learning, as it was difficult to convey the fact and its importance.
> However, because you gave us the idea of how to explain it, and because you and the other reviewers also placed importance on the relationship with Hebbian learning (as NeurIPS), we could work on a major improvement in explaining the relationship.
> Now we believe our revised manuscript thoroughly explains what makes our method Hebbian. (line 48-76 with Figure 1)
>
> > Comparison with previous methods
> > In lines 104-109 the authors try to explain why they’re not comparing their method...
>
> We agree with the comment. We revised the description (line 113) with a new Figure 5 for the comparison.
>
> > Overall reason behind the method
> > Section 3.1: I did not understand why quantile normalization helps. ...
>
> Thank you for the comment. We added new figures (1b and 2c, with consequence 3d, 4c) and descriptions about why quantile normalization helps (line 58-76, 227-239). We now feel that these explanations are useful not only for our method but also for other studies, and that these explanations considerably increase the value of reading this paper.
>
>
> > Lines 135-136: "the backbone DNN model is a very good model...
>
> We agree with this comment and have amended this part (line 143).
>
> > 153: what probability distributions?...
>
> Thank you for the comment. After taking into account this comment, we have changed the related description (especially we changed “W” to “w_ave”), which changes this "probability distributions" to "frequency distribution". We think the newly created Figure 1(b) is useful also in clarifying what kind of distribution we are dealing with.
>
> > 166: "The range of applicable models is yet unclear...
>
> We agree with the comment and have removed this sentence. As described below, we added a subsection “Limitations, applications, and potential negative societal impacts” (line 167) and replaced it with sentences that would be more important as you suggested.
>
> > 223-224: what’s good accuracy for practical uses? ...
>
> We agree with the comment and have deleted this expression.
>
>
> > Small corrections
> > Note: the paper contains a lot of grammatical errors. ...
>
> > Line 15 (abstract): performs "at a", not "a", practical-level accuracy?
>
> > Also next sentence: Can write as "DONE overcomes ...
>
> > 92: uses "a" classification
>
> > 93: aim"s"
>
> > 145: applies to, not apply into
>
> > 212-214: black and orange said twice
>
> We agree with all these comments and have revised the manuscript according to them. Thank you for the comments.
>
> > Why is the supplementary material just a corrected version of the main paper?
>
> We thought that the information that is not directly related will use the time of the reviewers, but now we add 3 supplementary figures in this revised submission.
>
>
> > Limitations:
> > The authors can mostly addressed the limitations of their work (although see below), ...
>
> > Checklist 1C: the described model can be easily adapted for face recognition. ...
>
> > Related, the authors refer to practical uses throughout the whole paper, ...
>
> We agree with these comments and have added and revised the descriptions in the revised manuscript. Thank you for the idea and suggestion.
>
> Finally, we greatly appreciate the time you spent reviewing our paper, which was definitely essential to improve the core of the paper.

---

> > ### Comment · Reviewer_j4P6 · 2022-08-06
> > **Response**
> >
> > Thank you for your response! I'm mostly satisfied with the small corrections, but I don't think my main concerns are addressed. So I'm leaving the same score (4, reject).
> >
> > 1. Comparison with other methods
> >
> > Looking at Fig. 5, the proposed method works basically as well as Qi's method, and worse than some other methods. There's some improvement wrt interference for EfficientNet (Fig. 3d), but I don't think that improvement alone (given exactly the same performance for ViT) justifies a NeurIPS paper.
> >
> > 2. Hebbian learning
> >
> > I've checked the update explanation, and I still think the method has nothing to do with Hebbian learning.
> >
> > The concept of Hebbian learning has a specific meaning in neuroscience (see Chapter 8 in Theoretical Neuroscience by Dayan and Abbott). The main idea is that synaptic changes should depend on activity of the pre- and post-synaptic neurons. Modifications of the standard Hebbian rule like BCM rule, or Oja's rule, or 3-factor Hebbian rules have their own names, but still emphasize the dependence on pre- and post-synaptic activity. In your case, Qi's rule (Eq. 2) is indeed Hebbian -- the post-synaptic neuron is fixed at 1, the pre-synaptic one is fixed at $x$, so the weight becomes $x$. I think that calling quantile normalization Hebbian is a big stretch -- yes, it depends on re-normalized inputs, but they're re-normalized according to a weight distribution which real neurons wouldn't have access to. This breaks any meaningful connections with Hebbian learning as a biological concept. One hopefully clarifying example is backprop: weight changes depend on the activations at the pre-synaptic layer and the propagated error. Despite the dependency on the pre-synaptic activations, it's not considered a Hebbian rule.
> >
> > Small correction in line 236: neuronal refers to actual neurons. For artificial neurons the usual word is "neural".

---

> > > ### Author Response · Authors · 2022-08-08
> > > **Thank you very much for reading our revised manuscript and for providing more focused and constructive discussion points.**
> > >
> > > Thank you very much for reading our revised manuscript and for providing more focused and constructive discussion points.
> > > We believe here we can reach a shared understanding through our response below, thanks to your clear comments again.
> > > > Comparison with other methods
> > >
> > > DONE is a method for class addition tasks, and accuracy in Fig. 5 is not for showing DONE's advantage but for comparing various backbone DNNs and for evaluating other methods that use optimizations (as written in the paper, and as you probably understand).
> > >
> > > On the other hand, interference in class addition is important for DONE as the class addition is the main task (Fig. 3d). Not only with EfficientNet, but also with most CNNs, a similar thing happens because the statistical features of x and wi vectors are different.
> > > (we will add the results in the supplemental material).
> > >
> > > Qi's method misrecognized 10% of the input images as belonging to the new classes, despite the chance level of approximately 1% (8/1008). The addition of new classes actively interfered with the original classification (10 times more than chance), which suggested the situations in which this method could be used were limited. On the other hand, DONE's interference was about the chance level (1%) even with EfficientNet.
> > >
> > > DONE and Qi's method showed almost the same results when the backbone DNN was ViT (as you commented, and the reason was also explained in the paper). However, as mentioned above, DONE and Qi's method would give different results in most current backbone DNNs except ViT. In addition, the future backbone DNNs could also make the difference. We believe that there is a big difference between DONE, which can be used non-parametrically for any DNN, and Qi's method, which can only be used with limited DNN such as ViT (just like the difference between parametric and non-parametric statistical tests).
> > >
> > > We believe the discussion here is constructive and we would like to express it in the paper, e.g., by adding a sentence like "Qi's method misrecognized 10% of the input images as belonging to the new classes despite the chance level of about 1% (8/1008), while DONE's interference was about the chance level (1%).” at line 265.
> > > > Hebbian learning
> > >
> > > Thank you again for the clear explanation. With our explanations below, we believe we can reach a shared understanding that our paper presents a new implementation of Hebbian theory. Before the discussion here, we would like to let you know that we think it would be worth changing "Hebbian weight imprinting" to "Quantile weight imprinting" to avoid confusion if we can change the title at the camera-ready submission or the subsequent submission (it was impossible to change the title at the rebuttal revision), although our paper is actually related to Hebbian theory as described below.
> > > > but they're re-normalized according to a weight distribution which real neurons wouldn't have access to.
> > >
> > > We think this is the key point, and thus we here focus on it. Real neurons do not need access to the information of weight distributions. Real neurons inevitably satisfy the characteristics of real synaptic strengths, as a physical constraint of x and y neurons. Our method implements this physical constraint of neurons by using quantile normalization. Hebbian theory inevitably involves such transformations (influences) from neural activity to synaptic strength, and our method provides a realistic implementation of this transformation.
> > >
> > > Conversely, for real neurons, it is impossible to have a linear relationship between neural activity and synaptic strength, although it would be one of the simplest implementations for calculations.
> > >
> > > Therefore, our paper proposes a more realistic implementation of the Hebbian theory. Moreover, our paper shows it works better also as a method, because quantile normalization is a non-parametric method and it can realize the above-mentioned realistic implementation with any backbone DNN.
> > >
> > > We again believe this discussion is constructive and would like to reflect it in the paper, e.g., by modifying a sentence at line 58 as " Here, a problem arises with this simple formulation alone, because neural activity and synaptic weight are different in scale and those relationships would not be linear, not only in real brain but also in DNN.” and adding a sentence like “Note that real neurons do not need access to the information of weight distributions, but they inevitably satisfy the characteristics of real synaptic strengths as a physical constraint of neurons.” at line 76.
> > > > One hopefully clarifying example is backprop: ...
> > >
> > > We never thought about the relationship between backpropagation and Hebbian theory. We have not yet quite figured it out, but we are excited about the idea. Thank you for giving us the interesting new idea.
> > > > Small correction in line 236: ...
> > >
> > > Thank you for the comment. We will correct it.
> > >
> > > We would like to express our gratitude again for your time and comments for the constructive discussion.

---

> > > > ### Comment · Reviewer_j4P6 · 2022-08-08
> > > > **Still disagree wrt results and Hebbian connection**
> > > >
> > > > 1. results
> > > > > Not only with EfficientNet, but also with most CNNs, a similar thing happens because the statistical features of x and wi vectors are different.
> > > >
> > > > Currently the interference results only have EfficientNet, however. Fig. 5 suggests that Inception will have a similar results due to lower performance of Qi's method, but not ResNet12. (Btw, I don't recall a single paper that used ResNet12. It's usually ResNet18 or 50.)
> > > >
> > > > I don't think you can claim that what you see for ViT and EfficientNet will happen for all transformers and all CNNs. I don't disagree that it's an interesting difference for these two architectures, and that it justifies the method. However, I consider it a minor improvement over Qi's method as currently it is only happening for one specific architecture.
> > > >
> > > > 2. Hebbian learning
> > > > > Real neurons do not need access to the information of weight distributions. Real neurons inevitably satisfy the characteristics of real synaptic strengths, as a physical constraint of x and y neurons. Our method implements this physical constraint of neurons by using quantile normalization.
> > > >
> > > > The last sentence contradicts the first, as you need weight information for quantile normalization wrt weights.
> > > >
> > > > > Therefore, our paper proposes a more realistic implementation of the Hebbian theory
> > > >
> > > > It doesn't. Nothing in the paper discusses how your idea relates to real neurons.
> > > >
> > > > Again, "uses pre-synaptic activities" doesn't mean "Hebbian". It has a more specific meaning used in neuroscience (see the explanation and the link to a book above), which I could not find in the paper.
> > > >
> > > > 3. Backprop
> > > > I gave an example of an algorithm that fits your definition of Hebbian as it uses pre-synaptic activity for weight updates, but is not considered Hebbian by the community.

---

> > > > > ### Author Response · Authors · 2022-08-08
> > > > > **Thank you very much for your prompt reply.**
> > > > >
> > > > > Thank you very much for your prompt reply.
> > > > >
> > > > > > 1. results
> > > > > > Currently the interference results only have EfficientNet, ...
> > > > >
> > > > > We are sorry that our previous response "Not only with EfficientNet..." was unclear. This description meant the results of the interference in Figure 3d (and supplementary results that will be added), not Figure 5. Figure 5 does not show the results of class addition and thus the interference is irrelevant, therefore the difference between DONE and Qi's method is not important.
> > > > >
> > > > > We used Resnet-12 because it was a standard model used in one-shot learning research. We will add the results with Resnet-50 and VGG16 in Figure 5 (no significant difference between DONE and Qi's method).
> > > > >
> > > > >
> > > > > > I don't think you can claim that what you see for ViT and EfficientNet will...
> > > > >
> > > > > We agree with the comment. We already have results on the differences in the distributions of x and w_i vectors that give rise to differences in DONE and Qi's method, also for other DNNs such as ViT-B16, ResNet50, VGG16, and MobileNetV2. All but ViT have right-tailed x distributions and bell-shaped w_i distributions, similar to EfficientNet.
> > > > >
> > > > > We will show those results in the supplemental material. We are sorry that we did not explain it to you properly in the previous response.
> > > > >
> > > > > Nevertheless, as you commented, we do not think we can claim that what we see for those DNNs will happen for all transformers and all CNNs, but we think it is much better than EfficientNet alone.
> > > > >
> > > > >
> > > > >
> > > > > > 2. Hebbian learning
> > > > > > The last sentence contradicts the first,...
> > > > >
> > > > > The first sentence is about real neurons, and the last sentence is about implementation in ANN, and thus they are not contradictory.
> > > > > This time we also refer to another method to explain that using the weight information at the implementation does not separate our method from the framework of Hebbian.
> > > > >
> > > > > For example, "synaptic saturation" is used in some Hebbian implementations.
> > > > > Real neurons would not have access to the information of the saturation. However, when implementing it in ANN, an arbitrary maximum value can be used as the saturation.
> > > > > This implementation can be interpreted as the physical constraint of neurons that is inevitably given to real neurons, and the method does not leave the framework of Hebbian.
> > > > >
> > > > > Our method does something similar. Our method uses weight information just to set the physical constraint of neurons, just like the synaptic saturation.
> > > > > From the above, the first and last sentences are not contradictory, and the use of the weight information in the implementation does not separate our method from Hebbian.
> > > > >
> > > > >
> > > > > > It doesn't. Nothing in the paper discusses how your idea relates to real neurons.
> > > > > > Again, "uses pre-synaptic activities" doesn't mean "Hebbian"...
> > > > > > 3. Backprop I gave an example of an algorithm ...
> > > > >
> > > > >
> > > > > Our method is a weight imprinting that does not modify W_ori (original W matrix), which is explained by pre- (x) and postsynaptic (y) firing.
> > > > > It is not like the Non-Hebbian form such as solely on the basis of presynaptic firing.
> > > > > We think we have already got your understanding that the explanation so far is related to Hebbian.
> > > > > Therefore, we think the problem you point out is quantile normalization.
> > > > > (By the way, Qi's method changes W_ori)
> > > > >
> > > > > In the book you introduced (we will cite it, thank you), there is the following description (page 281 in the book):
> > > > > "General forms of the Hebb rule state that synapses change in proportion to the correlation or covariance of the activities of the pre- and postsynaptic neurons."
> > > > > However, this narrow sense can not apply to the synaptic saturation, which are treated as within Hebbian in the book. Therefore, we do not think your comment is that this narrow sense diverges our method from Hebbian.
> > > > >
> > > > >
> > > > > In our method, just like the synaptic saturation, we transform xy to $\\Delta w$ by using quantile normalization as a computational formulation of the physical constraint of neurons.
> > > > > At the quantile normalization, the strength ranking of xy remains unchanged, and only the scale is changed. The scale transformation is nonlinear, like the synaptic saturation.
> > > > > Therefore, like the synaptic saturation, we do not think our method is far from Hebbian theory.
> > > > >
> > > > > Certainly, quantile normalization is a new process for Hebbian implementation, thus in that sense, it would be necessary to discuss whether it fits within the framework of Hebbian so far.
> > > > > Therefore, we believe this discussion is meaningful, and indeed our understanding has also improved, thanks to your comments.
> > > > > Now we do not think that our method departs from Hebbian as described above, and so far we have not seen a definition that indicates our method is far from Hebbian (except the narrow definition above).
> > > > >
> > > > >
> > > > > If you still think our above description is wrong, we would appreciate it if you could point out specifically where it is wrong again, as you did in the previous comments.
> > > > >
> > > > > Thank you very much again for your time and important comments.

---

> > > > > > ### Comment · Reviewer_j4P6 · 2022-08-09
> > > > > > **More Hebbian learning**
> > > > > >
> > > > > > > synaptic saturation
> > > > > >
> > > > > > It is a physical constraint on the size of the synapse (and the amount neurotransmitters that can fit in) that real neurons do have access to (even if implicitly).
> > > > > >
> > > > > > > However, this narrow sense can not apply to the synaptic saturation
> > > > > >
> > > > > > Why not? Synaptic changes can be proportional to some quantity until they reach a fixed limit. I don’t see a contradiction.
> > > > > >
> > > > > > > synaptic saturaion vs. quantile normalization
> > > > > >
> > > > > > Your algorithm is non-local, unlike synaptic saturation. Hebbian plasticity with synaptic saturation would only use activity of a single pre-synaptic neuron, a post-synaptic neuron, and the size of the synapse itself. Your algorithm would need information about _all_ pre-synaptic neurons and information about other weights in the network to first rank the pre-synaptic neurons, and then communicate the appropriate weights to individual synapses. So it has two issues for real neurons: weights from the rest of the network have to be messaged to the synapses at one particular neuron; pre-synaptic neurons have to be ranked according to that weight information. Both of these things can’t just happen at a neural level, as this is a non-trivial computation. So it’s not similar to synaptic saturation.
> > > > > >
> > > > > > > Definitions of Hebbian
> > > > > >
> > > > > > 1. The classic one from Hebb (https://neurology.mhmedical.com/content.aspx?bookid=3024&sectionid=254335819#1180645941):
> > > > > > > According to Hebb’s rule: “When an axon of cell A … excites cell B and repeatedly or persistently takes part in firing it, some growth process or metabolic change takes place in one or both cells so that A’s efficiency as one of the cells firing B is increased.” The key element of Hebb’s rule is the requirement for coincidence of pre- and postsynaptic firing, and so the rule has sometimes been rephrased as “Cells that fire together, wire together.”
> > > > > >
> > > > > > 2. Introduction of Chapter 8 of Dayan and Abbott:
> > > > > > > In this chapter we largely focus on activity-dependent synaptic plasticity
> > > > > > of the Hebbian type, meaning plasticity based on correlations of pre- and
> > > > > > postsynaptic firing. To ensure stability and to obtain interesting results, we
> > > > > > often must augment Hebbian plasticity with more global forms of synaptic
> > > > > > modification that, for example, scale the strengths of all the synapses onto
> > > > > > a given neuron. These can have a major impact on the outcome of develop-
> > > > > > ment or learning. Non-Hebbian forms of synaptic plasticity, such as those non-Hebbian
> > > > > > plasticitythat modify synaptic strengths solely on the basis of pre- or postsynaptic
> > > > > > firing, are likely to play important roles in homeostatic, developmental,
> > > > > > and learning processes.
> > > > > >
> > > > > > Note the “augment” part for other forms of plasticity, and the “correlation between pre and post”, not just pre-synaptic activity.
> > > > > >
> > > > > > 3. Nonlinear Hebbian learning https://journals.plos.org/ploscompbiol/article?id=10.1371/journal.pcbi.1005070
> > > > > >
> > > > > > — this one still uses $\mathbf{x} f(y)$ for a pre-synaptic vector $\mathbf{x}$ and post-synaptic scalar $y$. Note no information exchange between synapses.
> > > > > >
> > > > > > 4. Three-factor rules https://www.frontiersin.org/articles/10.3389/fncir.2015.00085/full#h3
> > > > > >
> > > > > > Eq. 1 formulates Hebbian learning as $\Delta w = H(pre, post)$ for _two_ neurons, so it doesn’t depend on the rest of the network. (As an aside, your rule could fit three-factor Hebbian form $H(M, pre, post)$ for modulation $M$, but it’d have to be synapse-specific. Which defeats the purpose of the third factor, so the rule stays as much Hebbian as backprop, i.e., not Hebbian.)
> > > > > >
> > > > > > The bottomline everywhere is pre- and post-synaptic activity for _two_ neurons, not a set of pre-synaptic neurons. Your algorithm needs all pre-synaptic neurons, and also external information, making it non-Hebbian.

---

> > > > > > > ### Author Response · Authors · 2022-08-09
> > > > > > > **Thank you so much for the specific reply. We have finally come to an understanding of your comment. We totally agree with it.**
> > > > > > >
> > > > > > > Thank you so much for the specific reply. We have finally come to an understanding of your comment. We totally agree with it. Our quantile normalization does not belong to Hebbian theory.
> > > > > > >
> > > > > > > > The bottomline everywhere is pre- and post-synaptic activity for two neurons, not a set of pre-synaptic neurons.
> > > > > > >
> > > > > > > This must be the key point. We finally understand your point.
> > > > > > >
> > > > > > > > we often must augment Hebbian plasticity with more global forms of synaptic modification
> > > > > > >
> > > > > > > From this description, our method is something beyond Hebbian. Now we clearly understand your earlier comment that meant "before quantile" was Hebbian but "quantile and after" was not.
> > > > > > >
> > > > > > > Our method is certainly Hebbian-inspired and includes Hebbian, but it is wrong to call quantile normalization "Hebbian weight imprinting".
> > > > > > >
> > > > > > > Therefore, the title and corresponding text should be changed to "quantile weight imprinting." Also, related statements such as "the implementation of Hebbian theory" should be revised. However, these changes are minor and do not affect the main point of our paper (we think as you suggested).
> > > > > > >
> > > > > > > Thus if our paper is accepted, we will email the program chairs that we want to change the title at the timing of camera-ready etc. (it seems to have been possible at least in 2021)
> > > > > > >
> > > > > > > As above, we have finally come to an understanding of your comment. We really appreciate your patient discussion to improve our paper when you must be very busy.
> > > > > > >
> > > > > > > Thank you from the bottom of my heart.

---

> ### Comment · Reviewer_j4P6 · 2022-08-09
> **Not Hebbian; score upped to 5**
>
> The authors have been convinced that the method is not Hebbian, and agreed to change the paper accordingly (see the thread under my initial review). They also clarified several parts regarding the performance of the method.
>
> I'm still hesitant to call it a clear accept as the contributions remain limited, but I appreciate the discussion we had, so I'm switching my score from borderline reject (4) to borderline accept (5).

---

> > ### Author Response · Authors · 2022-08-09
> > **We again sincerely appreciate your wonderful support in your limited time.**
> >
> > Thank you very much for not only your precious comments and discussion to improve our paper, but also for raising the score.
> >
> > We have learned a lot from the discussion with you. Thanks to you, as with our papers, our understanding also progresses, which will be encouraging for future research as well.
> >
> > We again sincerely appreciate your wonderful support in your limited time.
> > (We hope this reply is within the time limit, and please allow this brief/quick gratitude in comparison to your great contribution.)

---

### Official Review · Reviewer_RJBn · 2022-07-11

**Rating:** 6
**Confidence:** 4
**Soundness:** 3 good
**Presentation:** 3 good
**Contribution:** 3 good

**Summary:**

The authors present Direct ONE-shot learning with Hebbian imprinting (DONE) a method for one-shot learning inspired by Hebbian learning in the brain. The method uses neural activations from the final layer of an “encoder” network (such as a vision transformer or EfficientNet) on a single example from an unseen class to create a weight vector for the class of the new class. The presented method is closely related to and improves upon a that of a 2018 paper by Qi et al, which the authors cite.

The authors present experiments in which DONE is compared to Qi’s method. Specifically, the authors one-shot-learn 1 or 8 classes, evaluate the performance on those classes, and measure the degree to which the new classes interfere with initially trained classes. The authors also present results on k-shot learning.


**Questions:**

1. The authors write on line 31 that “The human brain does not necessarily have more complex processes than DNNs…”. I recommend to remove this statement. Many researchers would disagree with the statement, and the statement is not essential to the paper.

2. The authors write on line 33 that “a series of simple processes such as linear filtering followed by a nonlinearity can describe the function of lower visual cortex”. I recommend to rephrase this slightly, as learning in the lower visual cortex is part of the lower visual cortex’s function, but that aspect is not described by a series of simple processes such as linear filtering followed by a nonlinearity.

3. “modify” on line 48 should be “modifies”

4. On line 61, the authors write “neural activity and synaptic strength are different in dimension.” Do the authors mean  “neural activity and synaptic strength are different in scale.”? Line 157 also mentions “different dimension” when the authors may have meant “different scale”.

5. On line 135, the authors write “the backbone DNN model is a very good model as a heritage of mankind”. I did not understand this sentence. Could the authors rephrase it?

6. Figure 2 is impossible to understand without referring to explanations of data point markers in section 4.1, which means that a reader needs to jump back and forth between the Figure 2 and the text on the page after. I recommend explaining the figure markers in Figure 2’s caption instead of in section 4.1.

7. On line 185, the authors write “(not 1008 classes here)”. At this point, the reader has not yet read about the 1008-class experiments. How about removing the comment in parentheses?

8. The EfficientNet architecture is misspelled in a number of places in the paper, e.g. as “EfficinetNet” and “EfficientNnet”

9. Koray Kavukcuoglu’s name should be upper-cased on line 377.

**Limitations:**

The authors have adequately addressed the limitations and potential negative societal impact of their work.

**Strengths And Weaknesses:**

Strengths:
1. The paper presents a novel and simple method for few-shot learning.
2. While the paper does not claim to model the brain, it is exciting to see that one-shot learning with brain-line Hebbian imprinting can work so well.
3. This paper is a pleasure to read. It is well-structured, and the writing is mostly clear.

Weaknesses
1. The authors compare their method with only one method, when it would have been helpful for readers to compare to a broader range of existing one-shot learning approaches for image classification. The authors state “It is meaningless to compare the above three approaches with weight imprinting, because weight imprinting does not contain any optimization algorithm. “ . I disagree with this, for two reasons. First, papers such as the one about one-shot learning with siamese networks (https://www.cs.cmu.edu/~rsalakhu/papers/oneshot1.pdf), which the authors cited) are quite similar to DONE in that a big model gets trained on large amounts of data in a time-consuming process, and later one-shot classification becomes cheap. In other words, both were optimized at some point. Second, even if two models fall into different categories (e.g. because one is much more computationally expensive than the other), it’s useful for readers to know how much accuracy they lose (if any) by using a more flexible methodology.
2. While the algorithm from the paper is novel and undoubtedly very elegant, it is very similar to Qi’s method and other papers that the authors cite. In this sense, the paper is not a must-read for most researchers in the community.

---

> ### Author Response · Authors · 2022-08-02
> **Thank you for your encouraging and productive comments.**
>
>
> Thank you for your encouraging and productive comments and ideas. Your comments have dramatically improved our paper.
>
> We really appreciate the comments, in particular, your comments convinced us of the value of explaining Hebbian theory more, not decreasing the Hebbian-related description, as well as the many necessary comments to improve the manuscript.
>
> Below we provide point-by-point responses to all comments in "weaknesses" and "questions".
>
> > The authors compare their method with only one method, when it would have been helpful for readers to compare to a broader range of existing one-shot learning approaches for image classification. ...
>
> We agree with the comment. We revised the description (line 113) and showed those results in newly created Figure 5.
>
> > While the algorithm from the paper is novel and undoubtedly very elegant, it is very similar to Qi’s method and other papers that the authors cite. In this sense, the paper is not a must-read for most researchers in the community.
>
> Our paper provides a new mathematical formation for Hebbian theory and shows that it works. Thanks to your comments, now the paper includes better descriptions about the relationship with Hebbian theory and the new implementation for it, which is not in the previous paper. Now we can believe our paper is a must-read for most researchers in the community.
>
> > The authors write on line 31 that “The human brain does not necessarily have more complex processes than DNNs…”. I recommend to remove this statement. Many researchers would disagree with the statement, and the statement is not essential to the paper.
>
> We agree with the comment and have removed this sentence. We are rather glad that many researchers would consider that the brain has more complex features, because it will encourage our future work. (Although unrelated to this paper, we personally would like to challenge dynamic and complex features of the brain.)
>
> > The authors write on line 33 that “a series of simple processes such as linear filtering followed by a nonlinearity can describe the function of lower visual cortex”. I recommend to rephrase this slightly, as learning in the lower visual cortex is part of the lower visual cortex’s function, but that aspect is not described by a series of simple processes such as linear filtering followed by a nonlinearity.
>
> We agree with the comment and have modified the description (almost removed) and its place. (line 36)
>
> > “modify” on line 48 should be “modifies”
>
> Thank you for pointing out our mistake. It is corrected in the revised manuscript.
>
> > On line 61, the authors write “neural activity and synaptic strength are different in dimension.” Do the authors mean “neural activity and synaptic strength are different in scale.”? Line 157 also mentions “different dimension” when the authors may have meant “different scale”.
>
> The comment is exactly right. "Scale" must be a much better wording and we have modified it throughout the paper. Thank you for your understanding and improvement ideas.
>
> > On line 135, the authors write “the backbone DNN model is a very good model as a heritage of mankind”. I did not understand this sentence. Could the authors rephrase it?
>
> We agree with this comment and have amended this part (line 143).
>
> > Figure 2 is impossible to understand without referring to explanations of data point markers in section 4.1, which means that a reader needs to jump back and forth between the Figure 2 and the text on the page after. I recommend explaining the figure markers in Figure 2’s caption instead of in section 4.1.
>
> We agree with this comment and have added markers explanation in the figure caption/legend. We also applied the same revision to all the other figures. Thank you for your improvement ideas.
>
> > On line 185, the authors write “(not 1008 classes here)”. At this point, the reader has not yet read about the 1008-class experiments. How about removing the comment in parentheses?
>
> > The EfficientNet architecture is misspelled in a number of places in the paper, e.g. as “EfficinetNet” and “EfficientNnet”
>
> > Koray Kavukcuoglu’s name should be upper-cased on line 377.
>
> We agree with these comments and have revised the manuscript according to them.
>
> Finally, we would like to express our gratitude again for the opportunity to have your precious comments, which not only greatly contributed to the improvement of our paper but also strongly encouraged us.

---

### Meta-Review · Area_Chair_FA2U · 2022-08-29

**Recommendation:** Reject
**Confidence:** Certain

**Metareview:**

## Summary
Humans can learn a new task just from a couple of examples whereas often supervised deep learning models require lots of labeled samples to learn a task.  This paper proposes a one-shot learning method inspired by Hebbian learning by adding a new class to the output layer of the network with quantile normalization of the new inputs based on the weights of the last layer that corresponding to the other classes. The proposed approach uses features of a penultimate layer of an “encoder” network (for example EfficientNet). The paper presents results in the k-shot classification setting.

## Decision
This paper studies an important problem and introduces interesting ideas such as quantile normalisation into deep learning models.  Nevertheless the paper requires a revision as discussed with other reviewers and as a result it would benefit from another round of reviews. However, during the discussion period none of the reviewers were willing to nominate this paper for acceptance with confidence.

* *Reviewer j4P6* claimed that the authors claims regarding to connections to Hebbian algorithm is flawed and  during the discussion period authors agreed their algorithm is not Hebbian. It seems like this is the biggest problem reviewer j4P6 has with theispaper. The reviewer also complained that the paper only has a handwavy explanation of the algorithm and some limited evaluations. It’s a decent contribution overall, but in its current form it does not meet the bar of NeurIPS.
* *Reviewer 8B2d* thinks that this paper needs another revision. The method presented in the article is interesting, but the current manuscript evaluates it with only two models, and the results are better in just one of them. It is therefore not clear whether the improvement is anecdotal or general. Furthermore, the manuscript relates the work to Hebbian learning in the brain, which is irrelevant for presenting their work. The authors agreed to that point when discussing with reviewer j4P6. The reviewer thinks that the article is interesting, but the authors should revise the manuscript, remove the Hebbian analogy and focus on better evaluations, making their statements more sound.
* *Reviewer RJBn*’s score is a “6 - Weak Accept”, but fine with this paper getting rejected. After reading the reviews of other reviewers, RJBn agrees with the other reviewers that the connections drawn to Hebbian learning in the brain are weak, and that the evaluation could be better.


**Award:**

No

---

### Decision · Program_Chairs · 2022-09-14

Reject